# Heartbeat Sound Signal Classification Using Deep Learning

**DOI:** 10.3390/s19214819

**Published:** 2019-11-05

**Authors:** Ali Raza, Arif Mehmood, Saleem Ullah, Maqsood Ahmad, Gyu Sang Choi, Byung-Won On

**Affiliations:** 1Department of Computer Science, Khwaja Fareed University of Engineering and Information Technology, Rahim Yar Khan, Punjab 64200, Pakistan; meetaliraza@outlook.com (A.R.); arifnhmp@gmail.com (A.M.); maqsood.dba@gmail.com (M.A.); 2Department of Information and Communication Engineering, Yeungnam University, Gyeongsan 38542, Korea; 3Department of Software Convergence Engineering, Kunsan National University, Gunsan 54150, Korea; bwon@kunsan.ac.kr

**Keywords:** heart sound, classification, deep learning, RNN

## Abstract

Presently, most deaths are caused by heart disease. To overcome this situation, heartbeat sound analysis is a convenient way to diagnose heart disease. Heartbeat sound classification is still a challenging problem in heart sound segmentation and feature extraction. Dataset-B applied in this study that contains three categories Normal, Murmur and Extra-systole heartbeat sound. In the purposed framework, we remove the noise from the heartbeat sound signal by applying the band filter, After that we fixed the size of the sampling rate of each sound signal. Then we applied down-sampling techniques to get more discriminant features and reduce the dimension of the frame rate. However, it does not affect the results and also decreases the computational power and time. Then we applied a purposed model Recurrent Neural Network (RNN) that is based on Long Short-Term Memory (LSTM), Dropout, Dense and Softmax layer. As a result, the purposed method is more competitive compared to other methods.

## 1. Introduction

Before the 19th century, doctors tried to diagnose the disease by listening to the heartbeat sound directly from a patient’s chest. This “immediate auscultation” is a very unethical and nonscientific approach for doctors. Laennec invented a medical instrument called the ”stethoscope” in 1816 that is used in the medical field to listen to the heartbeat sound. This instrument is widely used in the medical field to diagnose heart disease [1]. Besides, heart failure and coronary heart disease are difficult to diagnose for inexperienced or non-clinical persons by using a stethoscope. Despite accurate auscultating, this requires a long-term practice and several years of experience in a clinic, which is difficult to obtain [2].

The heartbeat sound of a normal heart involves two sound S1 (lub) sound components associated with closing valves systole and S2 (dub) sound component associated with opening valves diastole [3]. Doctors can find heart diseases from listening to the heartbeat by using a stethoscope. A normal heartbeat sound has a clear pattern of “lub dub, dub lub”, with the time dub to lub is greater than the time from lub to dub and its rate is 60–100 beats per minute (lub and dub). A murmur heartbeat sound has a noise pattern whooshing, roaring, rumbling or turbulent fluid between lub to dub or dub to lub and symptoms of many heart disease. An extra-systole heartbeat sound has an out of rhythm pattern “lub-lub dub, lub dub-dub” that normally is found in adults and very common in children [4].

Heartbeat sound classification usually comprises three steps. The first step is a pre-processing that cleans the heartbeat signal by passing a band-pass filter to eliminate the noise. The second step is a feature extraction technique that transforms each heartbeat sound signal into a fixed-sized heartbeat sound signal. Then, the Down-sampling [5,6] technique is applied that reduces the heartbeat sound signal frames. However, it helps us to decrease the computation time of our system without affecting the performance of our algorithm performance. The last step is to choose a suitable classifier that extracted features and complete classification tasks. In machine learning, classification algorithms used to classify new data by learns from previous data. Classification algorithms used in this study are Decision Tree (DT), Random Forest (RF) and Linear Support Vector Classifier (LSVC). Deep learning is a subset of Machine Learning. The state-of-the-art methods used in Deep learning algorithms are Multi-Layer Perceptron (MLP) and Recurrent Neural Network (RNN).

The main framework of this study shown in Figure 1 and comprises three steps: pre-processing, feature extraction, and the classification model. This study used Dataset-B from PASCAL Classifying Heart Sounds Challenge [7]. Dataset-B collected from a clinical trial in hospitals using a digital stethoscope (DigiScope). The first step is pre-processing, which cleans the signal by eliminating the undesired frequency. Then, we applied the feature extraction technique, down-sampling [5,6], which decreases the heartbeat sampling rate from 50,000 to 782 frames (divide 8 × 8 times). Later we applied the classification algorithm, i.e., Decision Tree (DT), Random Forest (RF), Linear Support Vector Classifier (LSVC), Multi-Layer Perceptron (MLP) and Recurrent Neural Network (RNN) in which MLP and RNN are Deep Neural Network. The more efficient classifier is an RNN Deep Neural Network, which gives the highest accuracy as compared to the other classification algorithms DT, RF and LSVC.

## 2. Related Work

Many researchers used different techniques, such as segmentation, down-sampling, feature extraction and classification. These techniques are used to predict heart disease. Elsa Ferreria Gomes et al. [7] proposed a system in which they convert the heartbeat signal to segmentation by using Shannon energy also an algorithm used for peak detection. After segmentation, they applied the classification model J48 and MLP algorithm. Unfortunately, they succeed in challenge 1 (segmentation) and does not provide the answer for challenge 2 (classification). J48 and MLP do not provide a good result due to this challenge 2 are not successful.

Elsa Ferreira Gomes et al. [4] describe a methodology for classifying heart sound PASCAL Challenge. They used an algorithm that identified the S1 and S2 heart sound where S1 is lub and S2 is dub. First, they applied the decimate function of MATLAB on the original sound signal and applied a band-pass filter to remove the noise in these signals. After that, they applied the average Shannon energy that is useful to identify the peaks of the heart sound signal easily. They used an algorithm in which they find the maxima and minima points of the sound signal to accomplished the segmentation of heartbeat sound. They used the J48 and MLP algorithm to train the model that predicts the sound signal. These signals are Normal, Murmur, Extra-sound and Artifact.

Yineng Zheng et al. [8] purposed a novel feature for detecting abnormality of heart sound and the features include first and second heart sound (S1-S2EF), energy fraction of heart murmur (HMEF), max energy fraction of heart sound frequency sub-band (HSEFmax), the entropy of S1-S2EF (S1-S2sampen). After that, they have normalized the sound signal then decomposed by wavelet packet. They applied a support vector machine (SVM) for a small dataset that consists of 80 normal heart sound and 167 murmur heart sound. The performance of the purposed method is good because of a small dataset that consists of 2 categories.

Shi-Wen Deng et al. [9] present a framework for heart sound classification based on an auto-correlation feature without using segmentation. The auto-correlation feature extracted from sub-band coefficients of heart signal with Discrete Wavelet decomposition (DWT) after that auto-correlation are fused to get the final features than they applied Support Vector Machine (SVM) on these final features and in Dataset-B [7] they get the precision of Normal (77), Murmur (76) and extra-systole (50) which is not too good.

Wenjie Zhang et al. [10] purpose a method to extract the discriminative feature of heartbeat sound classification by using scaled spectrogram and tensor decomposition. The spectrograms detected the heart cycles of fixed size to extract the discriminative feature. Tensor decomposition is used to reduce the dimension to get a more discriminative feature. After that, they applied the Support Vector Machine (SVM) to these features. The SVM model that gets the normalized precision is 0.74.

Yaseen et al. [11] used multiple feature extraction techniques for heartbeat sound classification. They used an electronics stethoscopes device to get the digital recording of the heart sound called a phonocardiogram (PCG). Mel Frequency Cepstral Coefficient (MFCCs) and Discrete Wavelets Transform (DWT) are the features extraction techniques used in this study. Furthermore, they combine the MFCCs and DWT feature extraction to enhance the results. They used various classification algorithms in this study, such as Support Vector Machine (SVM), Deep Neural Network and K-Nearest Neighbor (KNN).

Many researchers usually involve three steps of heartbeat sound classification. The first step is heart sound segmentation that detects by amplitude threshold-based method [12,13,14] and probabilistic- based method [15,16]. The second step is feature extraction based on time [17], frequency [18] and time-frequency [10,17,19] and the last step is the classification model. The commonly classification model used is SVM [8,9,10] and MLP [7] as shown in Table 1.

Deep Neural Network is the most emerging field in data mining. Deep learning is the subset of machine learning that used different types of layers to perform different tasks on data. These tasks are object detection and voice recognition. Many researchers have worked on deep neural networks to perform classification on the audio dataset.

Shawn et al. [20] have worked on a large-scale audio dataset and proved that its results are good by using Convolutional Neural Network. Yong Xu et al. [21] presented a gated Convolutional Neural Network that won the 1st place in the large-scale weakly supervised sound event detection and Classification of Acoustic Scenes and Events (DCASE) 2017 challenge. Yan Xiong Li et al. [22] used the BLSTM Network on Acoustic Scenes to get a better result. Kele Xu et al. [23] purpose a novel ensemble-learning system consists of CNN that gets a superior classification performance.

Most of the author [20,21,23] used CNN that performs well on image classification also RNN [24] performs well on a series of data. Image data gives the data as a 2D array that consists of the pixel value. Audio data give the data as a 1D array that consists of frequency value. In this study, Dataset-B consists of heartbeat sound samples that converted into 1D arrays. So, that is why the RNN model is used instead of CNN.

## 3. Methodology

The main framework of this study is shown in Figure 2 and comprises three steps: pre-processing, feature extraction and the classification model. First of all, split the Dataset-B [7] into two sets with a ratio of 70 and 30 percent training and test data. Then we applied the pre-processing techniques that eliminate the noise by applying a band-pass filter. After that, we fixed the size of the heartbeat sound signal by converting sampling frames into a 50,000 frame for each heartbeat sound file. Then we applied Down-sampling techniques that reduce the sampling frame rate 50,000 to 782 frames of the heartbeat sound signal by using decimate techniques. Now the classification model is ready to be applied and we predicted the results of our model.

The main framework of heartbeat sound classification has divided into two sections. Sections (a) and (b) are the training and test phases shown in Figure 2. In the training phase, the data insert as a signal and label then applied the pre-processing technique on these training data and applied feature extraction technique based on data framing and down-sampling. In the last, we have trained the purposed model.

In the test phase, the data inserted as a signal in which applied different techniques. These techniques consist of pre-processing, data framing and down-sampling. In RNN classification, get the trained model from the training phase and predict the heartbeat sound signal. After that, evaluate the purposed model by using the evaluation parameter i.e.accuracy.

### 3.1. Data Collection

Dataset-B [7] used in this study for classifying heart sounds PASCAL challenge competition. This Dataset-B contains 461 samples subdivided into three groups i.e., Normal, Murmur and Extra-systole. Each sample of Dataset-B carries 4000 sampling frequency rates i.e., file 1 contains total sample 26,551 (6.6 s), file 2 contains total samples 38,532 (9.6 s). The Dataset-B is displayed in Table 2.

A normal heartbeat sounds like e.g., lub dub or dub lub, and a murmur heartbeat sound has a noise between dub to lub or lub to dub and extra-systole heart sound is out of beats e.g., lub-lub dub or lub dub-dub [4]. These heartbeats sound waves are shown in Figure 3.

Figure 3 shows the normal sound wave with a precise shape of lub dub that has no noise. In murmur sound wave, showed a noise between lub to dub or dub to lub and extra-systole has a different shape like a lub-lub dub or lub dub-dub as appearing in Extra-systole sound wave. Spectrogram of these shapes, as presented in Figure 4.

A spectrogram is a visual representation of the sound signal waves, as shown in Figure 4, that presents three types of spectrograms i.e., Normal, Murmur and Extra-systole waves. Green shades described the amplitude of a sound wave in a spectrogram. The spectrogram of a normal wave is a clear sequence of amplitude i.e., lub dub. In the murmur wave, it shows a noise sequence of amplitude that is greater than normal and extra-systole sound waves. In extra-systole wave, the amplitude of a sound wave is higher than the normal sound wave but smaller than the murmur sound wave.

### 3.2. Pre-Processing

The heartbeat sound signal x(i) takes a 4 kHz sampling frequency rate and then cleaned heartbeat sound signal x(i) with a band-pass, zero-phase filter to get rid of the noise. However, filter the 50–800 Hz frequency to get rid of the noise in the heartbeat sound signal.
(1)$$x^{\prime }\left[i\right]=f\left(x\left[i\right]\right).$$
where $x^{\prime }\left[i\right]$ is the filtered heartbeat sound wave signal.

### 3.3. Data Framing

The Dataset-B [7] consists of 461 sound files in which each file is a specific second period with a 4000 sampling (Frame) rate. This sampling rate is the frame values of a sound file within 1-s and gets the overall frames by multiply the sampling rate to the time of a sound file.
(2)totalframe=samplingrate×time

For example, if a sound file f1 has a 5-s time, then the total frame rate of this file can be calculated by this formula.
(3)f1=4000×5.5=22000

Due to the different sound file times, the frame rate of each file is not equal and do not apply any classification algorithm because of different features length. So, data framing is used to fix the sampling rate of each record file. In data framing, converts the sampling frequency of each record file to a 50,000 (12.5 s) frame rate as shown in Figure 5.

Figure 5 shows the sound samples with a precise time 3-s, 7-s and 25-s. Then we convert each sound sample to 12.5-s (50,000 frames) samples. The sound file of 3-s repeats itself until it reaches 12.5-s. The sound file of 25-s can store only 12.5-s file parts.

Then, we transform each sample of a sound file into a 50,000 (12.5-s) frame rate that converts each sound file into a fixed size. Dataset-B gets the same feature of each sound file after that applied machine learning classification algorithms.

### 3.4. Down-Sampling

Each sound sample of Dataset-B contains 50,000 frame rates. It is a highly challenging task, dealing with this vast amount of features on our GPU. Now, it takes very high power and time to complete this task. To overcome this situation, reduce the features of a sound file to perform tasks more quickly, in less time.

For this purpose, applied down-sampling techniques [5,6] helped to reduce the sound signal frequency with a minor effect on its performance. Down-sampling (decimate) is a signal reducing technique applied to reduce the features of sound files.

Furthermore, the decimate technique applied to our sound files and its sampling frame rate 50,000 converts into a 782 frame rate by applying a low-pass filter (divide 8×8).
(4)$$x^{\prime }\left[i\right]=\frac{x[i]}{max(|x[i]|)}$$
where $x^{\prime }\left[i\right]$ is the normalized signal that converted 50,000 frame rate into 782 frame rate of heartbeat sound wave as shown in Figure 6.

We reduce the frame rate of heartbeat sound signal waves using down-sampling to extract the more discriminative features and further use these discriminative features in the classification model. In the x-axis, the features of the sound signal wave convert 50,000 to 800 frames, as displayed in Figure 6. Now we have to apply a classification algorithm in which it takes less computational power and less time to complete the task.

### 3.5. Proposed Method

RNN (Recurrent Neural Network) architecture applied in this research that consists of various layers such that Input, LSTM, Dropout, Dense and Softmax layer. Figure 7 displays the comprehensive architecture of the RNN model.

Table 3 displays the dimension and operations of each layer using by the RNN model. Involving the feature extraction techniques, thus it took 782 features of each sound file that passed into an input layer with shape 1×782×1. In the LSTM layer, the 1×782×1 shape changes into this 1×782×64 shape and applied the dropout layer. In the second LSTM layer, the shape has changed into 1×1×32 then applied dropout layer in which shape has not changed.

In the dense layer, the shape has converted into 1×1×3 which helps to classify the dataset because the dataset consists of three category classes. In the second to last layer, we only need three neurons equals to categorize the class length. Then we applied a Softmax layer that is used in the last layer to get the probabilities of each sample.

#### 3.5.1. LSTM Layer

The Long Short-Term Memory [25,26,27] consists of a connected block known as memory block are a part of the Recurrent Neural Network (RNN) and used in the artificial neural network. Each memory block in LSTM contains three gates, input, output and forget that perform different operations read, write and reset function. More precisely, the input of the cell multiplied by the input gate. The output of the cell multiplied by the output gate and the previous cell value multiplied by the forget gate, as shown in the following equations.
(5)$$f_{t}=\sigma_{s}\left(w_{f}x_{t}+U_{f}h_{\left(t-1\right)}+b_{f}\right)$$
(6)$$i_{t}=\sigma_{s}\left(w_{i}x_{t}+U_{i}h_{\left(t-1\right)}+b_{i}\right)$$
(7)$$o_{t}=\sigma_{s}\left(w_{o}x_{t}+U_{o}h_{\left(t-1\right)}+b_{o}\right)$$
(8)$$c_{t}=f_{t}\times c_{\left(t-1\right)}+i_{t}\times \sigma_{h}\left(W_{c}x_{t}+U_{c}h_{\left(t-1\right)}+b_{c}\right)$$
(9)$$h_{t}=o_{t}\times \sigma_{h}\left(c_{t}\right)$$
where $f_{t}$ is the forget gate activation vector, $i_{t}$ is the input gate activation vector, $o_{t}$ is the output gate activation vector, $h_{t}$ is the hidden state vector, $c_{t}$ is the cell state vector, $\sigma_{s}$ is the sigmoid function, $\sigma_{h}$ is the hyperbolic function, *w* is the weight, *b* is the bias vector and the initial values of $c_{0}=0$ and $h_{0}=0$.

#### 3.5.2. Dropout Layer

Over-fitting is a big issue in the field of machine learning. It occurs when a classification algorithm trains data and gives satisfying results. Then applied the classification algorithm on test data that gives the unsatisfied result. This happens when two or more neurons detect the same results repeatedly [28] and needs to drop the neurons that affect the results.
(10)$$z_{i}^{L}=w_{i}^{L}y^{L}+b_{i}^{L}$$
where *L* is a hidden layer in which $L\in l^{1},l^{2},l^{3},\ldots ,l^{n}$, $z^{L}$ is the input layers, $y^{L}$ is the vector output, $w^{L}$ is the weights and $b^{L}$ is the biases.

#### 3.5.3. Dense Layer

The layer is a set of neurons that get input as a weight, perform some linear function then passes the output to its next layer. The dense layer is the most basic neural network in deep learning that is used to change the dimensions of a layer. In the dense layer, all the neurons connected to the input and output layer.
(11)X=f(Y×w+b)
where *X* is the output layer, *Y* is the input layer, *w* is the weight, *f* is the activation function and *b* is the bias vector [17].

#### 3.5.4. Softmax Layer

Softmax is an activation function that is extremely important in an artificial neural network and decides whether the neurons are active or not. Softmax is an effective way that handles multi-class classification problems in which output represents in categorical ways [29]. The main goal of Softmax is to highlight the maximum value in neurons. Assigned the maximum neuron weight is one and assigned other neurons weight is zero. The activation function of Softmax defined as
(12)$$S\left(y_{i}\right)=\frac{e^{yi}}{\Sigma_{k}e^{yk}},k=1,2,\ldots ,k$$
where *y* and *S* is the input and output. The Softmax function is used in the last layer of the neural network to obtain the probabilities of the category class of each input.

### 3.6. Evaluation Criteria

There is a different type of several measures to evaluate the performance of classifier i.e., Accuracy Precision, Recall, specificity, sensitivity and F-Measure [30]. The evolution criteria used in Dataset-B is accuracy. To measure the accuracy, we need the number of true-positive (TP), false-positive (FP), true-negative (TN) and false-negative (FN) values defined as
(13)$$A=\frac{TP+TN}{TP+TN+FP+FN}$$

## 4. Results and Discussion

The objective of this study is to classify different heartbeat signals automatically. These results give the preliminary diagnoses that help to determine further processes in medicine. The dataset applied in this study is Dataset-B [7] that comprises three categories i.e., Normal, Murmur and Extra-systole. First, converted our Dataset-B into two datasets that consist of two different times i.e., 27.8-s and 12.5-s. Each sample of 27.8-s datasets contains 111,468 frames and 12.5-s dataset contains 50,000 frames.

In this study classification algorithm applies to the Dataset-B. Decision Tree (DT) and Random Forest (RF) is the Machine learning algorithm applied to heartbeat sound dataset to classify the heartbeat diseases. Decision Tree is the earlier classification algorithm used in a diverse area of classification. Random forest is an ensemble learning method used for text classification and generating random decision tree [31]. These algorithms perform well on categorical and text datasets. Their performance on voice and image dataset is unsatisfied. As a result, DT and RF have gained less accuracy displayed in Table 4 as compare to the neural network.

Deep learning is an artificial neural network that acts like a human brain to learn and make a decision on its own. For classifying audio and image datasets, authors [20,21,32,33] prefers the deep learning algorithm and their performance very well. Deep learning is the subset of machine learning, as well as machine learning, which is a subset of artificial intelligence. In deep learning, data passes through each layer. That is the output of the previous layer and input to its next layer. The first and last layer is called the input and output layer. Another layer between the input and output layer is considered to be a hidden layer in which each layer performs an activation function. The deep learning algorithm extracts the discriminative feature itself but extracts the feature separately in the machine learning algorithm. The deep learning algorithm extracts the feature itself. However, machine learning needs to extracts the feature separately.

Multi-Layer Perception is a feed-forward network consisting of input, output and hidden layers. It trained by back-propagation where the weight connected with layer and mostly used for regression and classification problems [34]. It is the simple neural network that uses Dense, Dropout and Softmax layer. In this study, Multi-Layer Perceptron gained 67% and 69% accuracy by using 6 and 16 layers. Increasing the hidden layers in Multi-Layer Perceptron also increases the accuracy. But at some level, increasing the layers does not affect the accuracy anymore.

Recurrent Neural Network (RNN) gains the highest accuracy in our study based on Long Short Term Memory (LSTM), Dropout, Dense and Softmax layers. LSTM layer is used for time-series analysis and remember the past data in a memory cell. Each time step, a few gates are used to deal with the passing of information along the sequence that can capture long-range information more precisely [35,36]. This is the reason that our purposed model has gained the highest accuracy because it remembers the previous information for a long period that connects to the present task. Adam optimizer used in our RNN model that gives the best result compared to other optimizers Adagrad and SGD. The dropout layer is an effective way to avoid over-fitting because it randomly drops some of the connections between layers that help to prevent over-fitting [37]. Furthermore, our hyper-parameter model tuned here by applying the different dropout rates {0.05, 0.20, 0.35}. Each dropout rates applied to our RNN model that gives the different results shown in Table 5. Hence, it shows that increasing the dropout rate increases the accuracy and decreases the loss. 0.35 is the ideal dropout rate that gives the best results in our model.

During the training of the 12.5-s dataset of our purposed model, RNN used 322 samples in which 289 samples used for training and 33 samples used for test data.

The validation accuracy and loss in training shown in Figure 8 help to find out the model working is fine or not. If validation loss increases and validation accuracy decreases, that means our model is not in the learning state. If validation loss and accuracy are increases, that means our model is over-fitting. If validation loss decreases and validation accuracy increases, then it means model learning and working is fine. Figure 8 shows increases validation accuracy and decreases validation loss. It means our model learning is fine and gained the highest accuracy as compared to other classification algorithms.

Figure 9 shows the results in which the Recurrent Neural Network (RNN) gained the highest accuracy on both 12.5-s and 27.8-s sample files in Dataset-B. RNN gained 80.8 Accuracy with 12.5-s sample files and 77.2 Accuracy with 27.8-s sample files in Dataset-B. Decision Tree gained the lowest accuracy in both 12.5-s and 27.8-s sample files in Dataset-B. It also shows that the overall results of 12.5-s sample files are higher than the results of 27.8-s sample files. Recurrent Neural Network (RNN) is the overall best performing classification algorithm in our model.

K-Fold Cross-Validation is a re-sampling technique used to evaluate the classification algorithm on limited data samples. The result of K-Fold Cross-Validation is less biased or less optimistic compared to other methods, such as train/test split. K represents the number of folds is randomly partition of k equal-sized sub-samples. The 5-fold cross-validation is used in our classification algorithm to estimate the performance of the RNN model. Figure 10 shows the result of 5-fold cross-validation in which the RNN model gained the weighted accuracy and loss is 80.45 and 47.37.

The experimental results indicate that our purposed method has gained good results as compared to the previous work. Gomes et al. [7], Shi-Wen Deng et al. [9] and Wenjie Zhang et al. [10] gained the overall result on Dataset-B is 0.70 (MLP), 0.74 (SVM) and 0.76 (SVM). Our method has a good ability to extract the discriminative feature for heartbeat sound classification that gained the highest result is 80.8.

## 5. Conclusions

This research purposed a deep learning model for heartbeat sound classification based on data framing, Down-sampling, and RNN for Dataset-B. This purposed method can efficiently detect the heartbeat signal and gives information for deciding whether further treatment is necessary or not. Dataset-B divided into three categories Normal, Murmur and Extra-Systole. The heartbeat sound signal is cleaned by filtering the noise. The data framing converts the sampling frame rate of each audio file into a fixed-size frame rate than the down-sampling decrease the dimension of the heartbeat sound signal wave to extract the more discriminative features. The RNN proposed model was applied to the Dataset-B in this study that gained the highest accuracy 80.80. The experiment shows that our method is more competitive and efficient.

In future work, we will apply spectrogram techniques to get more discriminative features and applied other deep learning techniques like Convolutional Neural Network (CNN).

## Figures and Tables

**Figure 1 sensors-19-04819-f001:**
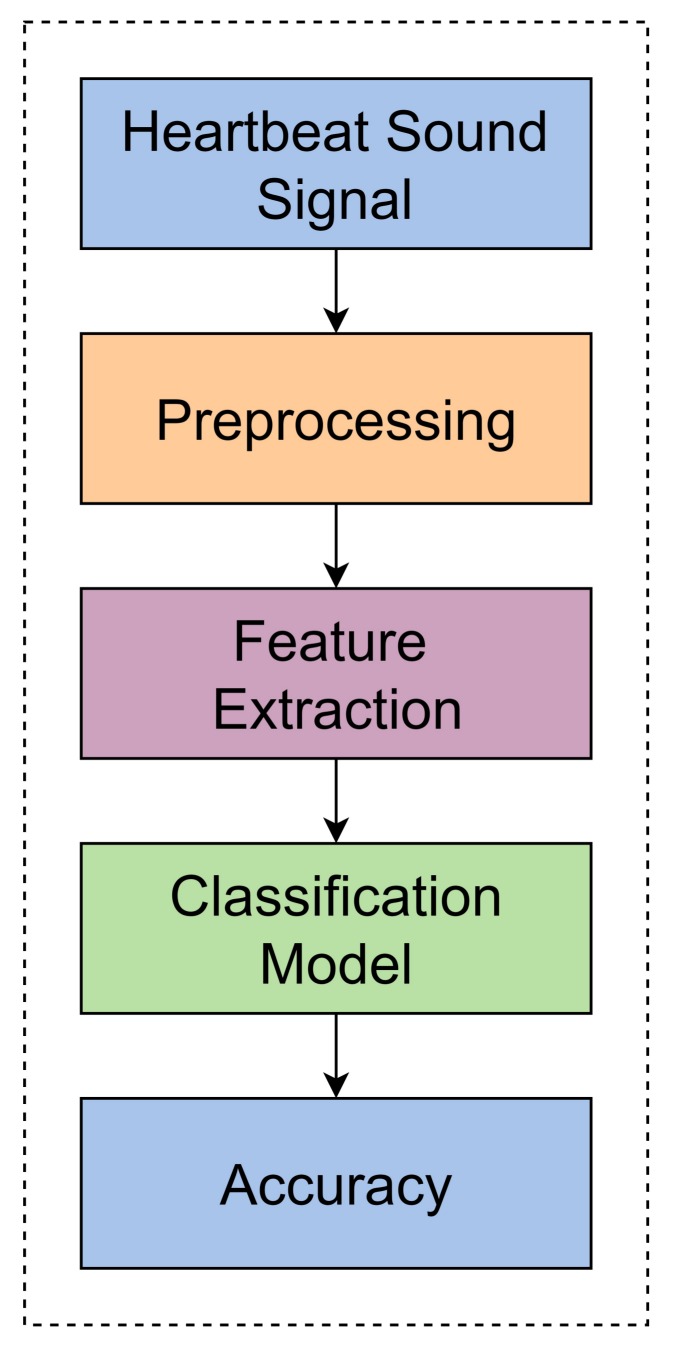
Architecture of this study.

**Figure 2 sensors-19-04819-f002:**
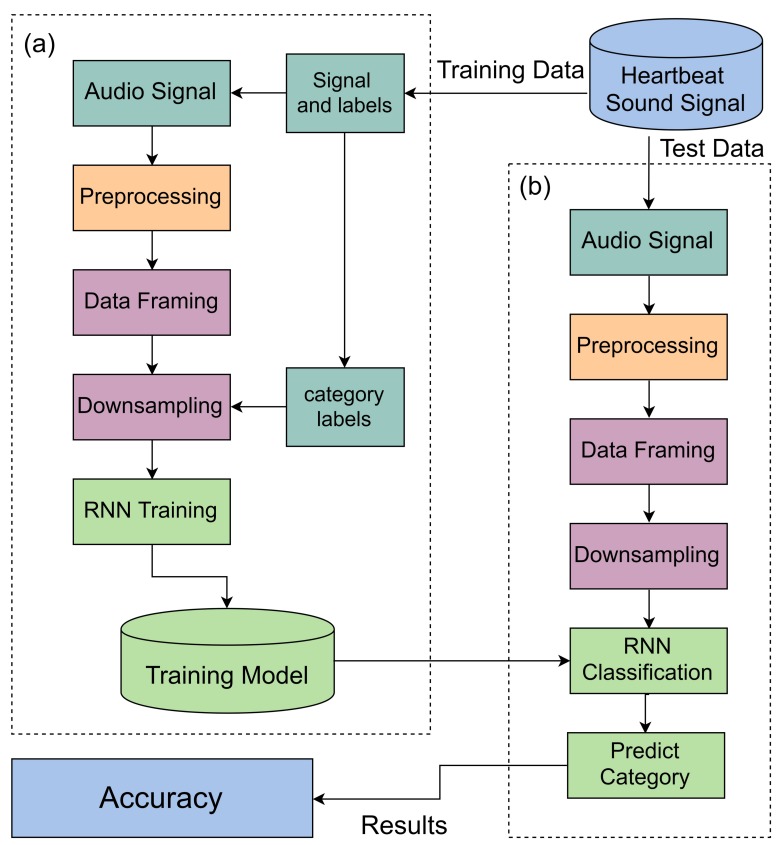
The main Framework of Heartbeat sound Classification, (**a**) is the training phase and (**b**) is the test phase.

**Figure 3 sensors-19-04819-f003:**
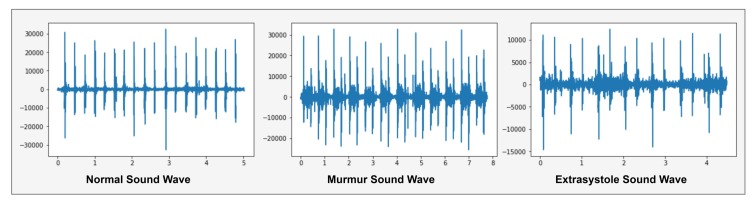
Graphical representation of Heartbeat sound signal.

**Figure 4 sensors-19-04819-f004:**

Spectrogram of Heartbeat sound signal.

**Figure 5 sensors-19-04819-f005:**
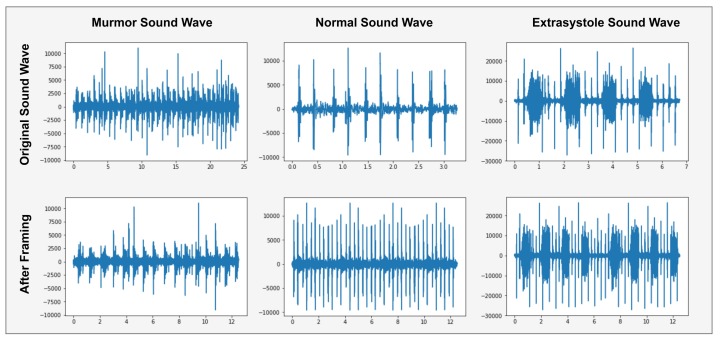
Graphical representation of Data Framing Heartbeat sound signal.

**Figure 6 sensors-19-04819-f006:**
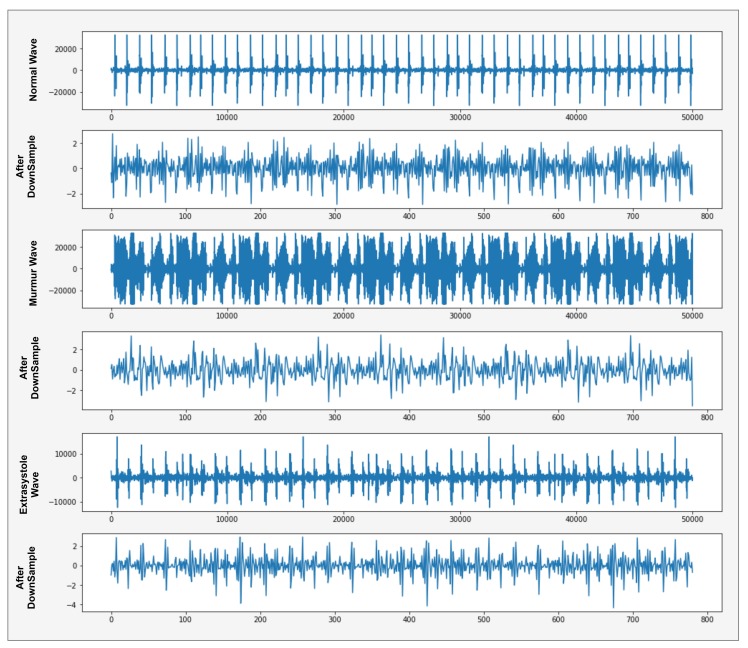
Graphical representation of Down-sampling Heartbeat sound signal.

**Figure 7 sensors-19-04819-f007:**
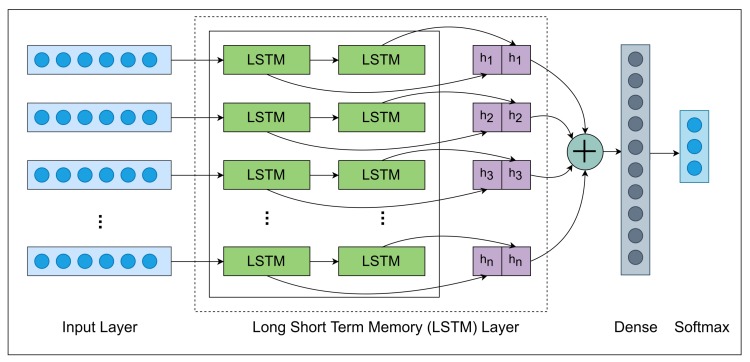
Overall Architecture of Recurrent Neural Network (RNN).

**Figure 8 sensors-19-04819-f008:**
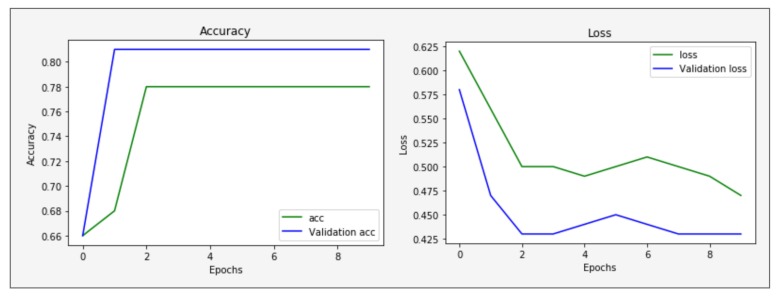
Validation accuracy and loss during the training of the RNN model.

**Figure 9 sensors-19-04819-f009:**
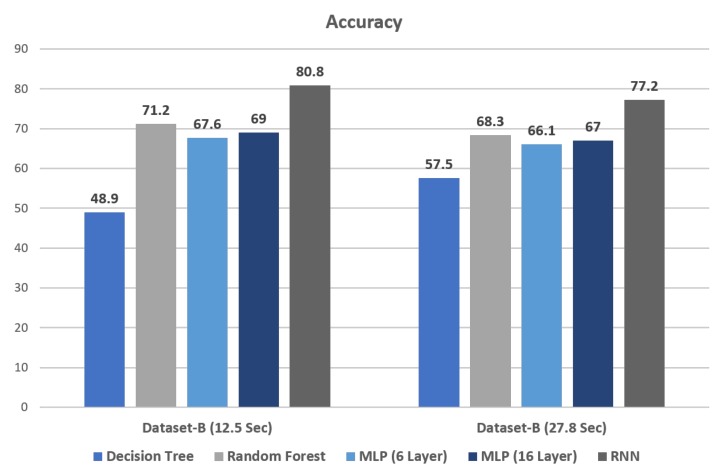
Accuracy Score comparison among 12.5-s and 27.8-s samples.

**Figure 10 sensors-19-04819-f010:**
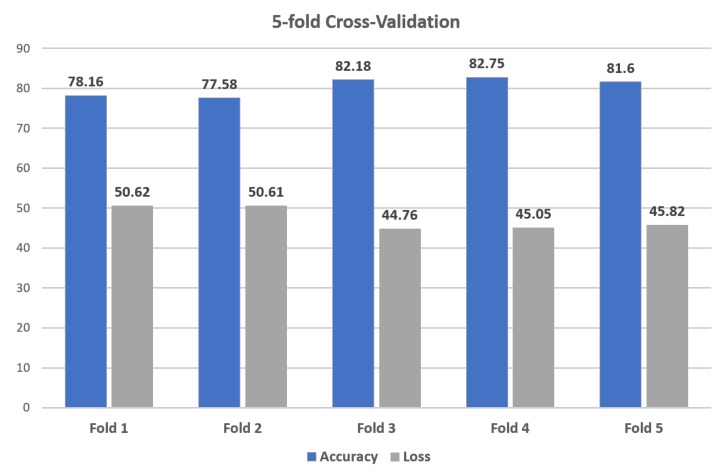
K-Fold Cross-Validation of 12.5-s samples.

**Table 1 sensors-19-04819-t001:** Summary of Previous Work on Dataset A & Dataset B.

Authors	Dataset	Model	Overall Results
[7]	Dataset A	MLP	2.12
[7]	Dataset B	MLP	1.67
[9]	Dataset A	SVM	0.74
[9]	Dataset B	SVM	2.03
[10]	Dataset A	SVM	0.76
[10]	Dataset B	SVM	0.74
[10]	Dataset A	2D-PCA	0.75
[10]	Dataset B	2D-PCA	0.68
[10]	Dataset A	MLP	0.59
[10]	Dataset B	MLP	0.62

**Table 2 sensors-19-04819-t002:** Dataset-B Detail.

Category	No of Samples
Normal	320
Murmur	95
Extra-systole	46

**Table 3 sensors-19-04819-t003:** Dimension and Operations of Layers.

Layer	Operator	Output Height	Output Width	Output Depth
Input	-	1	782	1
LSTM	-	1	782	64
Dropout	Rate = 0.35	1	782	64
LSTM	-	1	1	32
Dropout	Rate = 0.35	1	1	32
Dense	-	1	1	3
Softmax	-	1	1	3

**Table 4 sensors-19-04819-t004:** Overall Accuracy of 12.5-s and 27.8-s samples.

Classifiers	27.8-s	12.5-s
Decision Tree	57.5	48.9
Random Forest	68.3	71.2
Multi Layer Perceptron (MLP 6 Layer)	66.1	67.6
Multi Layer Perceptron (MLP 16 Layer)	67	69
Recurrent Neural Network (RNN)	77.2	80.8

**Table 5 sensors-19-04819-t005:** Hyper-parameter tuning of 12.5-s samples by Applying different Dropout rates.

Dropout Rates	Accuracy	Loss
0.05	76.10	52.30
0.20	77.93	49.59
0.35	80.81	47.05

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
