# Peer review of "Heartbeat Sound Signal Classification Using Deep Learning"

_sensors, 2019, doi:10.3390/s19214819_

Round 1
Reviewer 1 Report
1. Related Work
-I recommend that the authors organize more related things according to Figure 1.
- I also recommend that you clarify how the relevant work described is related or comparable to what is proposed in this paper.
2. Result and Discussion
- What fold-cross validation was executed?
- Because each category has different sized samples, the accuracy calculation seems to use weighted accuracy or F measure.
- It is necessary to discuss why the results of RNN performed better than other classifiers. However, I wonder if these classifiers are comparable to RNN.
- Why was the results not so high despite having only three categories.
3. Conclusion
- You said your method is competitive and efficient, but I can't agree that it's competitive. Extracting and using features may be inefficient, but I assume it may be better for performance. Can you defend against this?
4. References
- I recommend authors cite more "heartbeat sound classification" related references within the last three years(2017-2019). There are many old references.
Author Response
We would like to extend our deep gratitude and appreciation for the time and efforts that the editor and reviewers spent in reviewing this manuscript. We have tried to address all the comments and issues mentioned in the review report. We believe that the revised manuscript now meets the publication requirements and standards of the journal.
I attached my response file in which I respond to every reviewer's points one by one.

Reviewer 2 Report
This paper proposed an RNN-based heartbeat sound classification approach. Compared to the traditional shallow-architectures-based classifier, the proposed method provides competitive performance.
Generally speaking, the paper is short of novelty, which only employs the well-known deep network architectures for the classification task, and the contribution is very limited.
Moreover, I would suggest the author to polish this paper further, as there are many grammar errors and typos in this paper.
For example:
"Nowadays the majority of death caused by heart disease." Line 166, "3.5. purposed Method" Line 212, "3.6. Evolution Criteria" should be "Evaluation criteria"?
The audio classification task is an important task, which has drawn lots of attention during decades. Some reference can be helpful to refer to:
Hershey, Shawn, et al. "CNN architectures for large-scale audio classification." 2017 IEEE international conference on acoustics, speech and signal processing. IEEE, 2017.
Xu, Yong, et al. "Large-scale weakly supervised audio classification using gated convolutional neural network." 2018 IEEE International Conference on Acoustics, Speech and Signal Processing (ICASSP). IEEE, 2018.
Xu, Kele, et al. "General audio tagging with ensembling convolutional neural networks and statistical features." The Journal of the Acoustical Society of America 145.6 (2019): EL521-EL527.
Author Response

(The authors gave the same response as above.)

Reviewer 3 Report
This work proposes an algorithm for heartbeat sound classification using a machine learning framework. The methodology follows a standard paradigm, and ideally more technical novelty is expected from the authors. The idea is interesting, but there are several issues which need to be addressed before getting this work ready for publication:
Authors need to add more technical contribution to this paper, to make it ready for publication. For example exploring attention, or transformer network based idea can add more technical values. How are the model parameters and hyper-parameters tuned here? There should be a section in the experimental results showing the impact of varying each of these parameters on the final results. This work does not provide enough comparison with previous works (specially state-of-the-art deep learning based algorithms) for sound classification. Some of the statements are strange and needs to be corrected. E.g. “In machine learning, we used Decision Tree and Random Forest and in deep learning, we used230 Multi-Layer Perceptron and Recurrent Neural Network.”, all these models are a part of machine learning, including deep models. Perhaps authors refer to shallow vs deep architectures, so that need to be rephrased. There should be more explanation on the model architectural and why that is chosen. In terms of number of layers, and type of each one. Have the authors tried the CNN models directly on the spectrogram of these sounds? This is also one promising direction explored in several previous works. There are several grammatical errors which makes reading it very difficult. I recommend the authors to do a thorough proofread. Many of the recent works on sound classification, audio analysis, attention based models, are missing from the references. The authors need to do a more thorough literature study. Some of the relevant recent works on image segmentation are suggested below:
[a] "CNN architectures for large-scale audio classification." IEEE international conference on acoustics, speech and signal processing (ICASSP). IEEE, 2017.
[b] "Ad-Net: Audio-Visual Convolutional Neural Network for Advertisement Detection In Videos." arXiv preprint arXiv:1806.08612 (2018).
[c] "Large-scale weakly supervised audio classification using gated convolutional neural network." International Conference on Acoustics, Speech and Signal Processing, IEEE, 2018.
[d] "Deep-Emotion: Facial Expression Recognition Using Attentional Convolutional Network." arXiv preprint arXiv:1902.01019, 2019.
[e] “Acoustic Scene Classification Using Deep Audio Feature and BLSTM Network." International Conference on Audio, Language and Image Processing (ICALIP). IEEE, 2018.
Author Response

(The authors gave the same response as above.)

Round 2
Reviewer 1 Report
Result and Discussion
According to your answers I think you did not use fold-cross validation. In this case, data size is not so enough for training. Why don't you use 4 or 5 fold -cross validation?
conclusion
Why don't you add your "response 3" to the Related Work section. In addition, you need to refer and discuss the following works dealing with the other features in the Related Work section.
-Yaseen; Son, G.-Y.; Kwon, S. Classification of Heart Sound Signal Using Multiple Features. Appl. Sci. 2018, 8, 2344.
-Thiyagaraja, S.R.; Dantu, R.; Shrestha, P.L.; Chitnis, A.; Thompson, M.A.; Anumandla, P.T.; Sarma, T.; Dantu, S. A novel heart-mobile interface for detection and classification of heart sounds. Biomed. Signal Process. Control 2018, 45, 313–324.
Author Response
We would like to extend our deep gratitude and appreciation for the time and efforts that the editor and reviewers spent in reviewing this manuscript. We have tried to address all the comments and issues mentioned in the review report. We believe that the revised manuscript now meets the publication requirements and standards of the journal.

Reviewer 2 Report
I have no further comments as this paper is improved from many perspectives. Although the contribution seems to be limited, the study can be of interest to the readers of this journal. General speaking, I recommend the publication of this paper in this journal.
Author Response

(The authors gave the same response as above.)

Reviewer 3 Report
Thanks for addressing some of my concerns in the revised version. I think this work has improved over previous version, but still need several improvement to be ready for publication:
Authors need to add more technical contribution to this paper, to make it ready for publication. For example exploring attention, ensemble model, or transformer network based idea can add more technical values. The methodology used in the current version is very basic. How are the model parameters and hyper-parameters tuned here? There should be a section in the experimental results showing the impact of varying each of these parameters on the final results. This work does not provide enough comparison with previous works (specially state-of-the-art deep learning based algorithms) for sound classification. A lot more comparison are needed in the experimental results. By comparison I mean with previous works on standard benchmark, not just trying simple ML classifiers as shown in Table 4. Have the authors tried the CNN models directly on the spectrogram of these sounds? This is also one promising direction explored in several previous works. Captions of all figures should be self-explanatory. There are still several grammatical errors which makes reading it very difficult. I recommend the authors to ask a native speaker or someone with good writing style to proofread the entire paper. E.g. “The commonly used machine learning classifier is …DT, RF, …” should be “the commonly used classification algorithms are …”.
Author Response

(The authors gave the same response as above.)

Round 3
Reviewer 1 Report
This paper has been properly revised.
Author Response

(The authors gave the same response as above.)

Reviewer 3 Report
Thanks to the authors for addressing some of my concerns. The authors have explored LSTM for this classification task. They can also look at CNN, and the ensemble of CNN and LSTM, which can potentially improve this work. One such work present here:
"Deep-Sentiment: Sentiment Analysis Using Ensemble of CNN and Bi-LSTM Models." arXiv preprint arXiv:1904.04206, 2019.
There are still several issues which are not addressed completely by the authors, such as:
1- More comparison with previous works on some sounds classification benchmark.
2- Fixing grammatical errors and typos.
3- Discussing how the model parameters/hyperparameters are tuned and their impact on final performance.
I have raised this issue in previous round of reviews too, but have not seen any improvement in those directions. In any case, if they improve their English writing and more comparison with previous works, this work can be accepted.
Author Response
We would like to extend our deep gratitude and appreciation for the time and efforts that the editor and reviewers spent in reviewing this manuscript. We have tried to address all the comments and issues mentioned in the review report. We believe that the revised manuscript now meets the publication requirements and standards of the journal.

This manuscript is a resubmission of an earlier submission. The following is a list of the peer review reports and author responses from that submission.